# Body composition and physical activity as mediators in the relationship between socioeconomic status and blood pressure in young South African women: a structural equation model analysis

Richard J Munthali,[1] Mercy Manyema,[1,2] Rihlat Said-Mohamed,[1] Juliana Kagura,[1,3] Stephen Tollman,[4,5,6] Kathleen Kahn,[4,5,6] F Xavier Gómez-Olivé,[4] Lisa K Micklesfield,[1] David Dunger,[1,7] Shane A Norris[1]

For numbered affiliations see end of article.

**Correspondence to**
Dr Richard J Munthali;
munthali@aims.ac.za

## ABSTRACT

**Objectives** Varying hypertension prevalence across different socioeconomic strata within a population has been well reported. However, the causal factors and pathways across different settings are less clear, especially in sub-Saharan Africa. Therefore, this study aimed to compare blood pressure (BP) levels and investigate the extent to which socioeconomic status (SES) is associated with BP, in rural and urban South Africa women.

**Setting** Rural and urban South Africa.

**Design** Cross-sectional.

**Participants** Cross-sectional data on SES, total moderate and vigorous physical activity (MVPA), anthropometric and BP were collected on rural (n=509) and urban (n=510) young black women (18–23 years age). Pregnant and mentally or physically disabled women were excluded from the study.

**Results** The prevalence of combined overweight and obesity (46.5% vs 38.8%) and elevated BP (27.0% vs 9.3%) was higher in urban than rural women, respectively. Results from the structural equation modelling showed significant direct positive effects of body mass index (BMI) on systolic BP (SBP) in rural, urban and pooled datasets. Negative direct effects of SES on SBP and positive total effects of SES on SBP were observed in the rural and pooled datasets, respectively. In rural young women, SES had direct positive effects on BMI and was negatively associated with MVPA in urban and pooled analyses. BMI mediated the positive total effects association between SES and SBP in pooled analyses (ß 0.46; 95% CI 0.15 to 0.76).

**Conclusions** Though South Africa is undergoing nutritional and epidemiological transitions, the prevalence of elevated BP still varies between rural and urban young women. The association between SES and SBP varies considerably in economically diverse populations with BMI being the most significant mediator. There is a need to tailor prevention strategies to take into account optimising

## Strengths and limitations of this study

► The use of structural equation modelling allowed us to explore direct and indirect (mediation) effects of socioeconomic status, physical activity and body mass index on elevated blood pressure from a representative sample of rural and urban populations of South African young women.
► Although the urban and rural cohorts were from two different studies, the same research unit conducted both studies, and therefore, the data collection and management processes were consistent between the two sites, thereby allowing for accurate comparison.
► Other unmeasured data, such as undernutrition in infancy and dietary patterns, were not included in the current analyses.
► The low reliability of self-report data on physical activity could introduce bias. Thus, there is a need for more accurate, objective measures of physical activity to strengthen the results of our analysis.
► There is a need to do a comparison on longitudinal data, especially as the socioeconomic environment is changing rapidly due to rural–urban labour migration and other factors would be helpful to examine these associations over time.

BMI when designing strategies to reduce future risk of hypertension in young women.

## INTRODUCTION

High blood pressure (BP) or hypertension is a leading risk factor accounting for 7% of global disability-adjusted life years (DALYs) and contributing to the 34.5 million non-communicable disease (NCD) related deaths in 2010.[1 2] A recent global meta-analysis, involving 19.1 million individuals, reported that on average there has been a decrease in

high BP globally, but low-income to middle-income countries (LMICs) have seen an increase in hypertension.[3] The prevalence of high BP in LMICs is estimated at 30%[4 5] and it is the most significant risk factor for cardiovascular disease (CVD), most notably stroke.[6] In 2000, hypertension was estimated to have caused 9% of all deaths and over 390 000 DALYs in South Africa. Further, hypertension contributed to 50% of all strokes and 42% of ischaemic heart disease, signifying a substantial public health burden.[7] A systematic review of sub-Saharan African (SSA) data shows prevalence rates of hypertension of up to 41% with higher prevalence rates noted in urban compared with rural populations.[8 9] A study in men and women aged 40–60 years of age in six sites across four SSA countries, including South Africa, showed the same trend with South African urban and rural cohorts having the highest prevalence of hypertension (41.6%–54.1%).[10]

LMICs are experiencing both epidemiological and nutritional transitions with urban populations further along the transition as demonstrated by the higher prevalence of obesity and NCDs.[4 5 8 10–15] Some evidence has shown that there are differences in the levels of BP between rural and urban settings,[8] while other studies have found no significant differences.[16] According to Glass and McAtee, internal biological systems are sculpted by an interaction between genes and prolonged exposure to particular external environments, a principle they call embodiment.[17] Thus, the differences in built and social environments between rural and urban settings may explain the differences in disease prevalence. A Ghanaian study showed that both systolic BP (SBP) and diastolic BP (DBP) were significantly lower in rural participants compared with urban participants.[18] However, a similar study in adolescents found that BP levels were only lower in rural boys, with no difference in the girls.[19] Paediatric and adolescent hypertension have been reported to track into adulthood in a South African urban population.[20] Results on elevated BP from studies in rural South African children have reported prevalence rates varying from 1.0% to 25.4%.[21–24] The factors explaining these differences have not been fully studied in LMICs.

Socioeconomic factors such as education, household income and household assets have been associated with BP levels.[25–27] In a US cohort of young adults, a higher household income remained associated with lower SBP even after controlling for all potential covariates including age, sex and biobehavioural factors.[28] Similarly, in a French sample of 30–79 years, SBP independently increased and was inversely associated with both individual education and residential neighbourhood education.[29] Studies in African countries have also found varying associations between socioeconomic status (SES) and BP patterns, with both positive and negative associations reported.[8 30 31] Some studies have speculated that the association between SES and body mass index (BMI), physical activity (PA) levels, diet, smoking, alcohol intake and malnutrition may influence BP patterns.[18 28 31 32] PA has been inversely associated with BP and BMI directly associated with BP in

more advanced economies, but inconsistent associations have been reported in LMICs.[25 33–37]

There is a need to examine BP and its determinants in young South African adults given the high rates of overweight and obesity, and hypertension observed in this age group.[20 38] Recent South African reports also indicate that the highest pregnancy rates occur in the age range of 20–24 years, with 26.2% of births reported, followed closely by the age group of 25–29 years (25.7%),[39] and therefore, targeting young adult women would also reduce adverse health outcomes in their children. It is important to closely examine rural–urban differences in hypertension due to differences in the epidemiology of obesity and SES divergence in the South African context, in order to better suit interventions to the different settings.[23 26 30 40–43] Therefore, this study aims to compare BP levels between rural and urban young adult South African women, and to determine whether there is an association between SES and BP, and whether it is mediated by PA and BMI.

## METHODS

### Study sample and site

In the rural Agincourt site, 2016 potential female participants between the ages of 18 and 23 years were in the existing Agincourt Health and Socio-demographic Surveillance System database.[44] Only 996 were located during the data collection period and were invited to participate and of these, 509 female participants were recruited. The urban sample consisted of 510 young women between the ages of 22 and 23 years who were randomly selected from the sample of 720 females who were part of the Birth-to-Twenty plus (BT20+) Young Adult Survey.[45 46] Young women (n=51; 33 in rural and 18 in urban) who were pregnant at the time of the study were excluded, see the study design flow chart in figure 1. Measurements and questionnaires were completed by trained research assistants and nurses, and were standardised between both sites, to eliminate biases. Mentally or physically disabled women were excluded from the study.

### Patient and public involvement

No patients, private or public, were involved in this study, as it was a community population based.

### Blood pressure

BP (mm Hg) was the outcome variable and it was measured using an Omron 6 automated machine (Kyoto, Japan). A 5 min seated rest was observed before taking the BP measurements. Participants' seated BP was measured three times on the right side, with a 2 min interval between each measurement. The mean for the second and third readings was recorded for the current analysis. We had various cuff sizes and the appropriate size was used to accommodate differences in arm circumference.

According to the Seventh Report of the Joint National Committee on Prevention, Detection, Evaluation, and

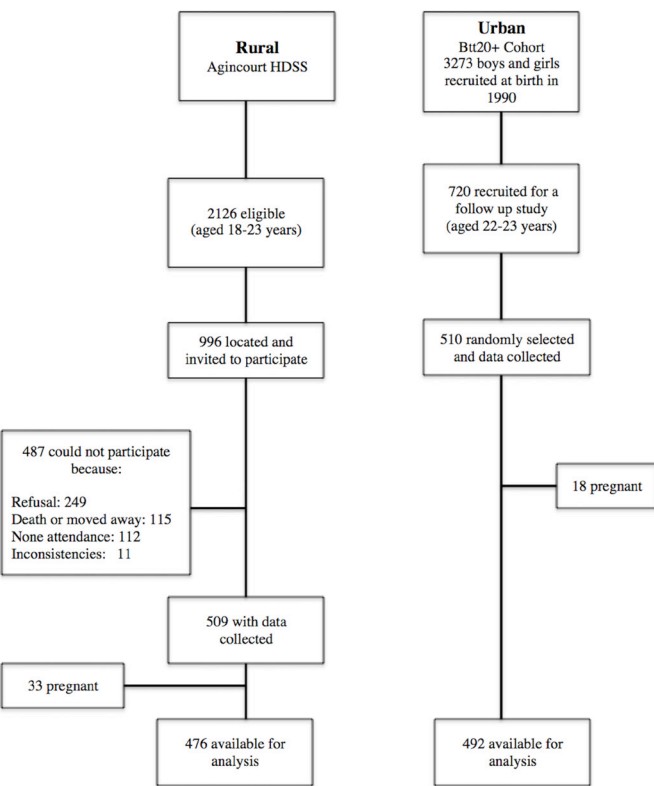

**Figure 1** Selection of study participants in rural and urban. Bt20+, Birth-to-Twenty plus; HDSS, Agincourt Health and Sociodemographic Surveillance System.

Treatment of High Blood Pressure,[47] five categories of BP have been established for adults 18 years of age and older as shown in table 1. These cut-offs were used in the current study. Prehypertension and hypertension were combined to create a new variable called elevated BP.

SBP was used in structural equation models (SEMs) as it is more relevant in adults, and a good predictor of adverse health outcomes later in life,[48] such as CVDs.

### Anthropometry
At both sites, participants' height and weight were measured by trained research assistants using standard techniques.[49 50] Weight was measured in light clothing and barefoot to the nearest 0.1 kg using a digital scale (Tanita model TBF-410; Arlington Heights, Illinois, USA). Height was measured barefoot to the nearest 0.1 cm using a stadiometer (Holtain, Crymych, UK). Waist circumference

**Table 1** Blood pressure classification[47]

| Classification | Systolic blood pressure | | Diastolic blood pressure |
|---|---|---|---|
| Low | <90 | Or | <60 |
| Normal | <120 | And | <80 |
| Prehypertension | 120–139 | Or | 80–90 |
| High: stage 1 hypertension | 140–159 | Or | 90–99 |
| High: stage 2 hypertension | ≥160 | Or | ≥100 |

(WC) was measured with a non-stretchable fibreglass tape at the level of the umbilicus. BMI was calculated as weight/height$^2$ (kg/m$^2$).

### Socioeconomic status
Physical assets owned in the participants' household were used as a proxy for SES index.[51] It was generated by summing the number of assets owned in the household from the following: television, car, washing machine, fridge, phone, radio, microwave, cell phone, Digital Versatile Disc (DVD)/video player, DSTV (cable channel), computer, internet, medical aid. Previous studies in this population have shown that the sum of physical assets (household assets) is closely related to the household per capital expenditure and household income.[51–53] The household SES is regarded as a good measure of accumulated household wealth so it is a more reflective wealth index than the income of a household's wealth over time.

### Physical activity
The Global Physical Activity Questionnaire, developed for global PA surveillance, was completed via interview to obtain self-reported PA.[54] Total moderate and vigorous physical activity (MVPA) in minutes per week (min/week) was calculated by adding occupation, travel-related and leisure time MVPA. Sitting time (min/week) was used as a proxy for sedentary time.

### Statistical analyses
Analysis of variance and Student's t-test, and χ$^2$ tests and Wilcoxon rank-sum test for non-parametric variables were conducted to compare study characteristics between urban and rural young women. SEM was used to test and estimate the direct and indirect associations between variables, most especially the mediation roles of PA (MVPA) or sedentary time (sitting) and body composition (BMI and WC), in the association between SES and SBP. SEMs allow us to assess the mediation effects of multiple mediators.[55] SEM decomposed SES-BP associations into two parts, direct (unmediated) and indirect (mediated through MVPA/sitting and BMI/WC).

Direct, indirect and total effects were computed and recorded, and the proportion of the total effect mediated was calculated. To evaluate the best fitting model for our data, we calculated different Goodness-of-Fit indices including χ$^2$ test, root mean squared error of approximation, Comparative Fit Index, Tucker-Lewis Index and standardised root mean squared residual.[56] Though the χ$^2$ test has been popularly used as a Goodness-of-Fit index, it has been reported to be biased and not reliable as the only Goodness-of-Fit index. It is also highly sensitive to sample size,[57 58] and often inflated with non-normal data such as PA data and we therefore employed the Hu and Bentler's Two-Index Presentation Strategy[56] combination rule, with cut-off values depending on the fitness index, to determine the best model fit.[56 59] We estimated the coefficients (β) with 95% CI for the direct, indirect and total effects and also calculated the proportion of association

**Table 2**  Descriptive characteristics

| | Total | n | Urban | n | Rural | P value |
|---|---|---|---|---|---|---|
| Age (years) | 22.04 (1.24) | 492 | 22.77 (0.49) | 476 | 21.28 (1.31) | 0.001 |
| Weight (kg) | 64.62 (14.82) | 492 | 64.67 (15.6) | 473 | 64.55 (14.03) | 0.9 |
| Height (m) | 1.61 (0.007) | 492 | 1.60 (0.07) | 475 | 1.61 (0.07) | 0.001 |
| BMI (kg/m$^2$) | 25.05 (5.59) | 492 | 25.32 (5.91) | 476 | 24.78 (5.24) | 0.13 |
| BMI classification (%) | | 492 | | 476 | | 0.015 |
| Underweight (<18.4 kg/m$^2$) | 5.98 | | 7.1 | | 4.82 | |
| Normal weight (18.5–24.9 kg/m$^2$) | 51.34 | | 46.45 | | 56.39 | |
| Overweight (25–29.9 kg/m$^2$) | 26.19 | | 29.21 | | 23.06 | |
| Obese (≥30 kg/m$^2$) | 16.49 | | 17.24 | | 15.72 | |
| WC (cm) | 80.60 (12.08) | 492 | 80.18 (12.63) | 476 | 81.03 (11.47) | 0.26 |
| Central obesity, WC ≥80 cm, % | 43.81 | 492 | 45.7 | 476 | 44.74 | 0.55 |
| Household SES index (sum of assets) | 7.24 (2.70) | 492 | 8.83 (2.37) | 476 | 5.59 (1.91) | <0.001 |
| Total MVPA (min/week)* | 870 (280–1810) | 492 | 420 (160–900) | 385 | 1680 (970–2580) | <0.001 |
| Sitting time (min/day)* | 300 (240–480) | 492 | 360 (240–480) | 385 | 300 (180–360) | <0.001 |
| Systolic BP | 106.68 (11.64) | 492 | 110.30 (11.4) | 471 | 102.89 (10.7) | <0.001 |
| Diastolic BP | 70.23 (9.00) | 492 | 72.78 (8.3) | 471 | 67.57 (9.0) | <0.001 |
| BP classification (%) | | 492 | | 471 | | <0.001 |
| Low BP | 12.46 | | 5.49 | | 19.75 | |
| Normal BP | 69.16 | | 67.48 | | 70.91 | |
| Prehypertension | 16.2 | | 23.58 | | 8.49 | |
| Hypertensive | 2.18 | | 3.46 | | 0.85 | |
| Elevated BP (%) | 18.38 | | 27.04 | | 9.34 | <0.001 |
| Highest education attained (%) | | 480 | | 371 | | <0.001 |
| Primary school | 1.18 | | 0 | | 2.7 | |
| Secondary school | 60.75 | | 48.33 | | 76.81 | |
| Tertiary education | 38.07 | | 51.67 | | 20.49 | |

Data presented as mean (SD) otherwise stated.

*Median (IQR).

BMI, body mass index; BP, blood pressure; MVPA, moderate and vigorous physical activity; SES, socioeconomic status; WC, waist circumference.

mediated by indirect effects. If the direct and indirect effects had opposite signs (negative or positive effects), the proportion mediated was assessed using the absolute values for all indirect and direct effects.[60]

All the analyses were conducted using STATA (V.13.0; STATA Corp). We confirmed SEM results by running the SEM with the Satorra-Bentler and Huber-White (Robust) Sandwich Estimator options[61] in STATA (V.15.1; STATA Corp). These options relax the normality assumption hence are robust to non-normal data, which would be the case for MVPA and SES in the current study. A p<0.05 was considered statistically significant.

## RESULTS
### Study characteristics
Descriptive statistics for the non-pregnant study participants (urban, n=492; rural, n=476) are presented in table 2. There was no difference in BMI or WC between the urban and rural participants, but the prevalence of overweight and obesity was significantly higher in the urban (46.5%) compared with the rural young women (38.8%). Household SES was significantly higher in the urban compared with the rural group. Self-reported MVPA was significantly higher in the rural than urban women (p<0.001), and the urban women spent significantly more time sitting than their rural counterparts (p<0.001). SBP and DBP were significantly higher in the urban group, as was the prevalence of elevated BP (27.0% vs 9.3%).

### SEMs for BMI and WC
Results from the SEMs for SES associations with SBP via MVPA and BMI are presented in table 3A–C for urban, rural and pooled analyses, respectively, and also shown in figures 1–3. No significant direct or indirect effects via (MVPA or BMI) of SES on SBP were observed in the urban women, but there were significant direct effects of SES on

**Table 3A** Structural equation model for SES, MVPA and BMI on SBP in urban women

| Effect of: n=489 | Outcome | Direct effects (95% CI) | Indirect effects (95% CI) | Total effects (95% CI) | Proportion of total effect mediated |
|---|---|---|---|---|---|
| Household assets | SBP via BMI | −0.34 (−0.75 to 0.07) | 0.05 (−0.05 to 0.14) | −0.29 (−0.70 to 0.12) | 0.13† |
| | BMI via MVPA | 0.13 (−0.09 to 0.35) | −0.014 (−0.05 to 0.013) | 0.11 (−0.11 to 0.33) | 0.1† |
| | MVPA | −41.71 (−73.48 to −9.94)** | | −41.71 (−73.48 to −9.94)** | |
| MVPA | SBP via BMI | −0.0002 (−0.001 to 0.001) | 0.0001 (−0.0001 to 0.0004) | −0.0000 (−0.0012 to 0.0011) | 0.3† |
| BMI | SBP | 0.37 (0.21 to 0.53)*** | | 0.37 (0.21 to 0.53)*** | |

Urban Fit Indices: LR test of model versus saturated: $\chi^2(4)=0.97$, probability $>\chi^2=0.91$; RMSEA=0.00; CFI=1.00, Comparative Fit Index. SRMR, 0.011: standardised root mean squared residual, CD=0.017, coefficient of determination; TLI, 1.12 Tucker-Lewis Index.
Adjusted for age; *P<0.05; **P<0.01; ***P<0.001.
†Assessed using the absolute values for both indirect and direct effects.
BMI, body mass index; LR test, likelihood ratio test; MVPA, moderate and vigorous intensity physical activity; RMSEA, root mean squared error of approximation; SBP, systolic blood pressure; SES, socioeconomic status.

MVPA. Results showed that individuals with a higher SES index were less likely to be physically active in pooled data and urban women. In rural women, a one-unit increase in total household assets was associated with a decrease of 0.65 mm Hg (95% CI −1.19 to −0.10) in SBP and an increase of $0.27 kg/m^2$ in BMI (95% CI 0.1 to 0.53) (table 3A,B and figures 2 and 3). The SEM for the pooled sample showed a significant indirect effect of household SES on SBP via BMI, with 50% of the total effect being mediated by BMI (table 3C and figure 4). Direct positive effects of BMI on SBP were observed in both settings and the pooled sample with a $1 kg/m^2$ increase in BMI being associated with an increase of 0.37 mm Hg (95% CI 0.21 to 0.53) and 0.33 (95% CI 0.12 to 0.54) mm Hg SBP in urban and rural young women, respectively. Similar results were observed when including waist circumference as the body composition indicator as shown in the SEM path diagrams with estimates in figure S1 (online supplementary data). The results from the SEMs with the Satorra-Bentler adjustment option, accounting for non-normality of the exposure, are shown in figure S2 (online supplementary data).

## DISCUSSION

A rising prevalence of hypertension has been reported in South Africa. Peer *et al* reported a higher prevalence in 2008 (35.6%) compared with 1990 (21.6 %) in men and women aged 25–74 years in an urban black community in Cape Town, South Africa.[40] We have shown in young adult women from urban and rural South Africa, an overall elevated BP prevalence of 18.4% (27.0% in urban and 9.3% in rural). We have also shown a direct effect of BMI on SBP in the urban and rural women separately, as well as when pooled, thereby providing further evidence of an association between overall adiposity and BP. The total effects of SES on SBP were the same in both settings.

Prevalence data on elevated BP and hypertension from other countries in SSA have shown conflicting results when comparing urban and rural communities. In Malawi, a higher prevalence of hypertension in urban compared with rural communities has been reported and attributed to differences in lifestyle as rural communities participate in subsistence-based agricultural activities while the urban community has a more westernised lifestyle with higher salt intake and lower physical activity.[9] Similarly, data from Ghana have shown a higher mean SBP and DBP and a higher prevalence of hypertension in urban communities.[18 62] In the Prospective Urban and Rural Epidemiological (PURE) study in South Africa, Pisa *et al* reported that both urban adult men and women had higher mean BPs in comparison to their rural peers though the overall CVD risk factors were equally prevalent in both settings.[41] In contrast, findings from Cameroon have reported a higher BP prevalence in rural compared with urban men and women older than 40 years old, while Kenyan studies have reported no significant differences.[16 63] Results from six urban and rural sites in four SSA countries—Kenya, South Africa, Ghana and Burkina Faso—have reported a prevalence of hypertension in women aged between 40 and 60 years ranging from 15.1% in rural Burkina Faso to 54.1% in urban South Africa.[10] It was also reported that in all the three South African sites, both rural and urban, the prevalence of hypertension was higher than in the other three countries.[10] These findings show the complex health transitions occurring in SSA and the impact that this is having on cardiometabolic disease risk.

Our study showed significant differences in SES between the urban and rural samples, as well a big variation in SES within these two settings. The social patterning of CVD risk factors, including hypertension, in SSA and LMICs, has in part been attributed to differences in countries' socioeconomic development. Previous results from five

**Table 3B** Structural equation model for SES, MVPA and BMI on SBP in rural women

| Effect of: n=378 | Outcome | Direct effects (95% CI) | Indirect effects (95% CI) | Total effects (95% CI) | Proportion of total effect mediated |
|---|---|---|---|---|---|
| Household assets | SBP via BMI | −0.65 (−1.19 to −0.096)* | 0.08 (−0.04 to 0.19) | −0.56 (−1.12 to −0.02)* | 0.11† |
| | BMI via MVPA | 0.27 (0.01 to 0.53)* | −0.01 (−0.04 to 0.01) | 0.26 (−0.005 to 0.53)* | 0.04 |
| | MVPA | −29.51 (−87.81 to 28.78) | | −29.51 (−87.81 to 28.78) | |
| MVPA | SBP via BMI | 0.0004 (−0.0005729 to 0. 0013) | 0.0001 (−0.0000 to 0.0003) | 0.0005 (−0.0005 to 0.0015) | 0.2 |
| BMI | SBP | 0.33 (0.12 to 0.54)** | | 0.33 (0.12 to 0.54)** | |

Rural Fit Indices: LR test of model versus saturated: $\chi^2$ (4)=10.51, probability >$\chi^2$=0.03; RMSEA=0.066; CFI=0.72, Comparative Fit Index, SRMR, 0.04: standardised root mean squared residual, CD=0.03, coefficient of determination; TLI, 0.37 Tucker-Lewis Index.
Adjusted for age; *P<0.05; **P<0.01; ***P<0.001.
†Assessed using the absolute values for both indirect and direct effects.
BMI, body mass index; LR test, likelihood ratio test; MVPA, moderate and vigorous intensity physical activity; RMSEA, root mean squared error of approximation; SBP, systolic blood pressure; SES, socioeconomic status.

**Table 3C** Structural equation model for SES, MVPA and BMI on SBP in the pooled sample of urban and rural women

| Effect of: n=867 | Outcome | Direct effects (95% CI) | Indirect effects (95% CI) | Total effects (95% CI) | Proportion of total effect mediated |
|---|---|---|---|---|---|
| Household assets | SBP via BMI | 0.23 (−0.08 to 0.54) | 0.23 (0.10 to 0.35)*** | 0.46 (0.15 to 0.76)** | 0.5 |
| | BMI via MVPA | 0.20 (0.05 to 0.34)** | −0.05 (−0.100 to 0.003) | 0.15 (0.01 to 0.29)* | 0.25† |
| | MVPA | −144.83 (−170.55 to −119.12)*** | | −144.83 (−170.55 to −119.12)*** | |
| MVPA | SBP via BMI | −0.001 (−0.002 to −0.0005)** | 0.0001 (−0.0000 to 0.0002) | −0.001 (−0.002 to −0.0003)** | 0.1† |
| BMI | SBP | 0.35 (0.21 to 0.49)*** | | 0.35 (0.21 to 0.49)*** | |

Pooled Fit Indices: LR test of model versus saturated: $\chi^2$ (4)=24.829, probability >$\chi^2$=0.000; RMSEA=0.077; CFI=0.89, Comparative Fit Index; TLI=0.75, Tucker-Lewis Index; SRMR=0.033: standardised root mean squared residual, CD=0.137, coefficient of determination.
Adjusted for age; *P<0.05; **P<0.01; ***P<0.001.
†Assessed using the absolute values for both indirect and direct effects.
BMI, body mass index; LR test, likelihood ratio test; MVPA, moderate and vigorous intensity physical activity; RMSEA, root mean squared error of approximation; SBP, systolic blood pressure; SES, socioeconomic status.

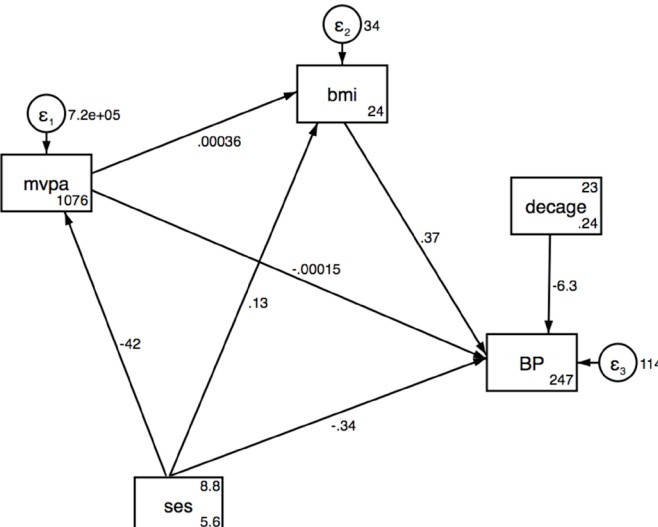

**Figure 2** Structural equation model for SES, MVPA and BMI on SBP in urban. BMI, body mass index; MVPA, moderate and vigorous physical activity; SBP, systolic blood pressure; SES, socioeconomic status.

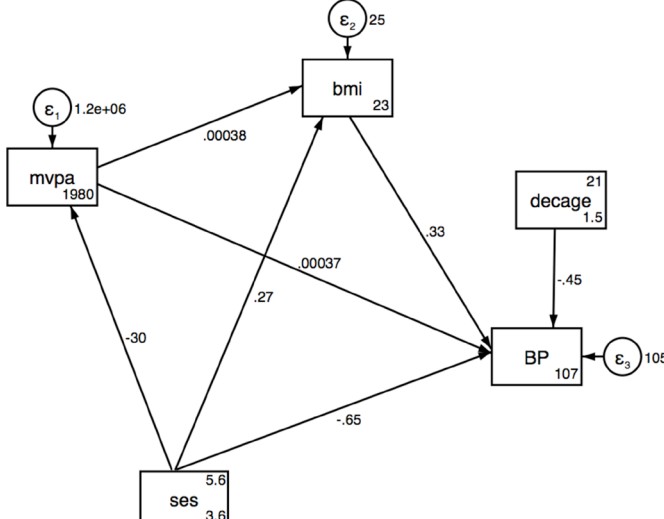

**Figure 3** Structural equation model for SES, MVPA and BMI on SBP in rural. BMI, body mass index; MVPA, moderate and vigorous physical activity; SBP, systolic blood pressure; SES, socioeconomic status.

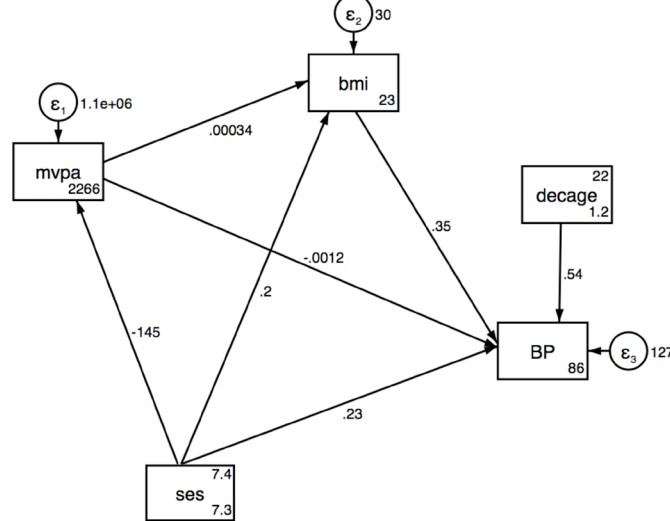

**Figure 4** Structural equation model for SES, MVPA and BMI on SBP pooled. BMI, body mass index; MVPA, moderate and vigorous physical activity; SBP, systolic blood pressure; SES, socioeconomic status.

countries (two high-income and three LMICs) reported that hypertension and other CVD risk factors were substantially associated with education and wealth status; individuals with less education and lower wealth generally showing a higher prevalence of CVD risk factors.[64] The effect of SES in this study is most evident in the rural women for whom household SES was lower (compared with urban) and who may be transitioning faster (both nutritionally and economically) than the urban young women. Though SES is positively associated with BMI in rural young women, it is negatively associated with SBP. There may be other factors, such as PA due to agricultural activities or dietary patterns, which were not recorded. In addition, the weight gain observed might not be due to

fat mass, which has been reported to be positively associated with SBP before,[65] but rather to muscle mass and bone mass.

In Mexico, women in rural and upper SES categories were likely to have a higher SBP, while we have reported that a higher SES was associated with a decrease in SBP in rural communities. At a population level, there is a need to consider different SES categories and monitor the effects of transitioning from one SES category to another on hypertension, since these categories may respond differently to an increase or a decrease in their SES. Kagura *et al* tracked SES in South African children and reported that moving from the low SES in infancy to a higher SES in adolescence had a protective effect on SBP level in young adulthood.[26] Our results have shown that this could be more pronounced in rural areas.

We observed a positive association between SES and BMI in the rural sample and the same direction of effects was observed in the urban, though not significant. This is in line with results reported in many LMICs including South Africa, but in contrast with those reported in higher income populations.[33 34 64] A systematic review of studies between 1989 and 2007 reported that SES was positively associated with obesity in the middle transitioning economies such as South Africa and Jamaica.[66] We have shown that both in the rural and urban participants (not significant), a higher SES resulted in reduced SBP, while the pooled analysis showed a positive total effect association between SES and SBP. This could be due to the introduction of more variation in SES when data from both sites are pooled; with many individuals with low SES in the rural area, the associations became skewed towards the low SES individuals. This may suggest that different transitional levels of SES have different effects on hypertension risk depending on the environment (either urban or rural). Though not significant, the total effects of SES on SBP

are the same in both rural and urban hence the differences in prevalence cannot be explained by the setting or SES alone. In urban and rural settings of four countries (Kenya, Namibia, Nigeria and Tanzania), the prevalence of age-standardised hypertension was similarly high and ranging from 19.3% to 38.0%.[11] Cois and Ehrlich reported that a higher SES was associated with lower SBP in a nationally representative sample of South African women[25] using SEM models. Alcohol use, PA, smoking and resting heart rate, and BMI were reported to be the mediators of the indirect of the association between SES and SBP in men but not in women, suggesting that other factors may play a major role in women.[25] Similarly, our results show that neither PA nor BMI mediates the association between SES and SBP in urban and rural settings in isolation, suggesting that other factors may explain the association. Among those, dietary patterns and stress have been reported to be independently associated with SBP.[67 68]

The significant direct associations between BMI and SBP are in line with other findings in South Africa and within the SSA region.[11 33 40 42 69 70] This link was consistent in rural, urban and pooled datasets, indicating the importance of BMI in the aetiology of high BP. Munthali *et al* reported that the link between obesity and hypertension could be observed as early as 5 years of age. Children with early onset of obesity were at higher risk of developing hypertension in late adolescence.[38]

In this study, using SEM models to explore the mediation role of BMI and PA helped quantify potential contributions of these variables to the effect of SES on SBP. The results show that PA was not a significant mediator in the association between SES and BP in the urban or the rural samples. SES was negatively associated with MVPA in urban and pooled samples, indicating that as individuals transition from low to higher SES, they reduce their PA level. We speculate that these differences in the association between SES and SBP in both our rural and urban results, and in those from high-income countries are due to differences in levels of nutritional and epidemiological transition in these regions.[71 72] Those with low SES in high-income countries are likely to consume cheaper, more energy-dense foods, participate in less leisure time PA and be more sedentary.[73 74] In LMICs, agricultural activities remain a part of everyday life and a day-to-day activity in rural living, while those with higher SES in the same settings rapidly adopt the westernised lifestyle with less PA, fewer agricultural activities and home-grown food. However, this speculation is not supported by the data on PA in this study despite the rural participants having a higher PA. Our understanding of the Agincourt rural economy is that agriculture is quite a minor aspect though very useful to augment the household income.

The limitations of this study are that other unmeasured data, such as undernutrition in infancy, which is a known risk factor for high BP later in life,[75] and dietary patterns were not included in the current analyses. We are currently working on research to address this limitation.

We can also not rule out the role of genetics. Second, the low reliability of self-report data on PA could introduce bias. Thus, there is a need for more precise, objective measures of PA to strengthen the results of our analysis. Lastly, longitudinal data, especially as the socioeconomic environment is changing rapidly due to rural–urban labour migration and other factors would be helpful to examine these associations over time. The cross-sectional design lacks a temporal component between the factors analysed. Thus, it is difficult to say anything certain about the direction of the associations, hence the need for the longitudinal data.

## CONCLUSIONS

Though the prevalence of overweight or obesity is relatively higher in both rural and urban than those reported in other SSA countries, women in the urban setting were at more risk for elevated BP than their rural counterparts. The link between SES and SBP varies in a more economically diverse population, as seen with the pooled rural and urban dataset, with BMI being the most likely mediator. There is a need to consider optimising BMI as a key intervention strategy in young adults in part to combat hypertension. Our findings should be replicated with prospective data.

**Author affiliations**
[1]MRC/Wits Developmental Pathways for Health Research Unit, Department of Paediatrics, School of Clinical Medicine, Faculty of Health Sciences, University of the Witwatersrand, Johannesburg, South Africa
[2]DST-NRF Centre of Excellence in Human Development, University of the Witwatersrand, Johannesburg, South Africa
[3]Division of Epidemiology and Biostatistics, School of Public Health, University of the Witwatersrand, Johannesburg, South Africa.
[4]MRC/Wits Rural Public Health and Health Transitions Research Unit, School of Public Health, Faculty of Health Sciences, University of the Witwatersrand, Johannesburg, South Africa
[5]INDEPTH Network, Accra, Ghana
[6]Umeå Centre for Global Health Research, Umeå, Sweden
[7]Department of Paediatrics, MRL Wellcome Trust-MRC Institute of Metabolic Science, NIHR Cambridge Comprehensive Biomedical Research Centre, University of Cambridge, Cambridge, UK

**Acknowledgements** We wish to thank the Bt20+ and Agincourt participants for taking part in the study and the Bt20+ and Agincourt team for their relentless support throughout the study.

**Contributors** RJM and SAN conceptualised the manuscript. RJM analysed the data. RJM, MM, RS-M, JK, ST, KK, FXG-O, LKM, DD and SAN interpreted the data. RJM wrote the manuscript and all authors were involved in editing and approving the final manuscript.

**Funding** SAN is supported by the UK MRC DfID African Research Leader Scheme and by the DST-NRF Centre of Excellence in Human Development at the University of the Witwatersrand, Johannesburg. Birth to Twenty data collection was supported by the Wellcome Trust under grant (092097/Z/10/Z). The MRC/Wits- Agincourt Unit is supported by the South African Medical Research Council, and the Wellcome Trust under grants (058893/Z/99/A, 069683/Z/02/Z, 085477/Z/08/Z, 085477/B/08/Z).

**Disclaimer** Opinions expressed and conclusions arrived at are those of the authors and are not to be attributed to the CoE in Human Development.

**Competing interests** None declared.

**Patient consent** Not required.

**Ethics approval** The study protocols were approved by the Human Research Ethics Committee of the University of the Witwatersrand (Clearance certificates M120138 for the Ntshembo-Hope Cross-Sectional Survey in Agincourt and M111182 for the BT20 + survey).

**Provenance and peer review** Not commissioned; externally peer reviewed.

**Data sharing statement** The datasets used and/or analysed during the current study are available from the Developmental Pathways for Health Research Unit data management department by contacting SAN on reasonable request.

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
