## [Reviewer comments · BMJ Open]

ARTICLE DETAILS

TITLE (PROVISIONAL)	Body composition and physical activity as mediators in the relationship between socio-economic status and blood pressure in young South African women: A structural equation model analysis
AUTHORS	Munthali, Richard; Manyema, Mercy; Said-Mohamed, Rihlat; Kagura, Juliana; Tollman, Stephen; Kahn, Kathleen; Gómez-Olivé, F. Xavier; Micklesfield, Lisa; Dunger, David; Norris, Shane

VERSION 1 – REVIEW

REVIEWER	Jo S Stenehjem Cancer Registry of Norway - Institute of population-based cancer research
REVIEW RETURNED	29-Apr-2018

GENERAL COMMENTS	COMMENTS TO THE AUTHORS Recap: This study aims to compare blood pressure (BP) between rural and urban young adult South African women, and to determine whether there is an association between socioeconomic status (SES) and blood pressure (BP) and whether it is mediated physical activity and BMI. The study is based on interview-assisted self-reports and clinical measurements in rural and urban black women (18-23 years age) using a cross-sectional design (i.e. measured without a temporal component between data). Comparison of blood pressure between rural and urban black women is stated as the outcome measure, and explanatory factors are listed as SES, body mass index (BMI), and moderate to vigorous intensity physical activity (MVPA). Results from their structural equation modelling showed significant direct positive effects of BMI and systolic BP (SBP) in rural, urban and combined datasets, and they conclude that the association between SES and SBP is mediated by BMI. Overall comment: 1: The authors present an important topic using an interesting approach with structural equation modelling (SEM) to estimate direct and indirect associations between known risk factors and BP. My main concern is that the limitations of the cross sectional study design are not adequately discussed. When measuring factors that are associated with each other without a temporal component between them, it is by design difficult to understand the direction of the associations. Even though some directions are more biologically plausible than other, you may end up with reverse causality (i.e. the outcome affecting the exposure). Also, the concept of mediation-analysis and use and interpretation of SEM might still be unfamiliar to many scientists, so maybe some
---

more detail and explanation is needed to reach a broader readership; VanderWeele TJ has several insightful publications worth citing f.ex: Annu Rev Public Health. 2016;37:17-32. doi: 10.1146/annurev-publhealth-032315-021402. Moreover, I miss a more precise description of how the final numbers entering the analysis of the two study populations were arrived at in the methods sections, possibly aided by a flow-chart to clearly show the inclusions and exclusions made. There are several inconsistencies in the use of abbreviations, and the manuscript would benefit from a close proof-read. Having said this, I think the data merit publication, but I recommend a revision of the paper that addresses the points above. I wish the authors good luck and would be happy to review it again. Detailed comments follow below.

Abstract:

2: Objectives: Abbreviate blood pressure; put BP in parentheses.

3: Design: Cross-sectional instead of cross-section

4: Results, line3: should it be "...effect of BMI on SBP".

5: Results: Please write "systolic BP (SBP)" for proper abbreviation.

6: Results, line6: use "BMI" (already abbreviated above)

7: Conclusions, line2: use "SES" (already abbreviated above).

Strengths and limitations bullet points:

8: Strength 1: Add "equation" between "structural" and "modelling".

9: Strength 2: When harmonization is listed as a strength I wonder what kind of harmonization. Could the authors specify exactly what data that were harmonized?

10: Limitation1: Suggest to delete "We are currently working on research..." from the bullet points as this is also mentioned in the main text.

11: Limitation2: I suggest to exchange "precise" with "valid" or "accurate" as biases are not threatening precision, but validity/accuracy.

12: Limitation3: Add "a" between "is" and "need" ?

Introduction:

13: p5, para1: Abbreviate "blood pressure" to "blood pressure (BP)" at first mentioning or use just "blood pressure" throughout the ms.

14: p5, para3: please define "SES" before abbreviation.

Methods:

15: p6, study sample: It is unclear whether pregnant young women (n=51) were subtracted before or after you arrived at n=509 for the rural sample and n=510 for the urban sample. It is also unclear how many of the young pregnant women who were rural and how many who were urban? Also, it is unclear how many who were excluded due to illness (last sentence). As mentioned in comment 1, I suggest to add a flow chart as a fig1 to clearly show the inclusions/exclusions made for both the rural and urban samples.

16: p6, blood pressure: Use of too large/small cuff-size has been associated erroneous BP readings. Were different cuff-sizes used to accommodate individual differences in overarm circumference?

17: p7, SES: Please write "SES" in first sentence (already abbreviated in intro)

18: p8, statistical analyses: Please use "MVPA" (already defined)

19: p8, statistical analyses: define and use "SBP" since you do not report DBP.

	20: p8, statistical analyses: As mentioned in comment 1, consider to give some more information on the interpretation of SEM – what do you estimate; β with 95% confidence intervals? Please also report significance level – 0.05? Results: 21: p8, study characteristics: I now understand that n=492 and n=476 were the numbers arrived at after excluding the 51 pregnant. However, I think it would be easier to follow with a flow chart as mentioned in comments 1 and 15. 22: p8, study characteristics: Just use MVPA (already defined) 23: p9: In the heading you write out Structural equation models, but abbreviate BMI – perhaps use full wording in headings and abbreviations in text after first mention for consistency? 24: p9, Structural equation models for BMI and waist circumference: Add “95% CI” when reporting from SEMs. Discussion: 25: p9, para2: use “SSA” instead of sub-Saharan Africa – already abbreviated. 26: p10, para2, last sentence: Please report whether the direction of the association for muscle mass and bone mass on SBP was positive or negative in ref #63. 27: p10, para3: Add “a” between “At” and “population”. 28: p10, para3: Be consistent in the use of “et al.” or “and colleagues”, and “pooled” or “combined” – choose one expression and stick with it throughout the ms as this is important for readability and clarity. 29: p11, para1: use either PA or physical activity consistently – in para4 “physical activity” is used again. 30: p11, para2: here “combined” and not “pooled” is used, and “et al.” instead of “and colleagues”.... 31: p11, para3: here “pooled” and not “combined” is used... Please excuse my hang-up, but I think it will improve the ms with more consistent use of expressions and abbreviations. 32: p11, para4, last sentence: As mentioned in comment 1, it is important that you elaborate here on the limitations inherent with a cross-sectional design; that you by design lack a temporal component between the factors analyzed. Thus, it is difficult to say anything certain about the direction of the associations. Conclusions: 33: p11-12: Based on comment 1 and 32 with respect to the limitations of the design, more prudence should be shown in the conclusions. You may consider to add something like: “Although our findings should be replicated with prospective data”
--	--

REVIEWER	Séverine Sabia Inserm, France
REVIEW RETURNED	18-May-2018

GENERAL COMMENTS	This paper reports social disparities in blood pressure among rural and urban young women from South-Africa. It raises an interesting question, particularly in the context of transitions occurring in this country, and the authors nicely discuss their findings in relation to this context. Analyses conducted separately among rural and urban women show interesting results both on SES disparities and on the associations between MVPA, BMI and blood pressure. However,
--

	results based on pooled analyses show confusing conclusion compared to stratified analyses. Is it possible that urban/rural environment is a missing confounding factors in the pooled analysis ? As household, BMI, physical activity and blood pressure differ strongly between urban and rural population, it would be nice to introduce this parameter into the model combining both populations. Higher physical activity is associated with lower blood pressure in the combined analysis but not in the stratified analysis. The distribution of physical activity in the pooled sample is likely to be bimodal according to table 1. How do SEM deal with such distribution ? Similar comment can be done for household assets. It would be nice to show association with waist circumference instead of BMI in appendix. Indeed in Table 1, we can see that rural participants have a lower BMI but a higher waist circumference (not significant). Showing in table 1, categories of waist circumference would be informative as well. It would be interesting to see association with alternative SES markers such as educational level. Is this information available ? The paper includes limitations that are well acknowledged in the discussion section, including the absence of information on diet patterns, the cross-sectional design and self-report of physical activity.
--	---

VERSION 1 – AUTHOR RESPONSE

Reviewer 1

Overall comment:

1: The authors present an important topic using an interesting approach with structural equation modelling (SEM) to estimate direct and indirect associations between known risk factors and BP. My main concern is that the limitations of the cross sectional study design are not adequately discussed. When measuring factors that are associated with each other without a temporal component between them, it is by design difficult to understand the direction of the associations. Even though some directions are more biologically plausible than other, you may end up with reverse causality (i.e. the outcome affecting the exposure). Also, the concept of mediation-analysis and use and interpretation of SEM might still be unfamiliar to many scientists, so maybe some more detail and explanation is needed to reach a broader readership; VanderWeele TJ has several insightful publications worth citing f.ex: Annu Rev Public Health. 2016;37:17-32. doi: 10.1146/annurev-publhealth-032315-021402. Moreover, I miss a more precise description of how the final numbers entering the analysis of the two study populations were arrived at in the methods sections, possibly aided by a flow-chart to clearly show the inclusions and exclusions made. There are several inconsistencies in the use of abbreviations, and the manuscript would benefit from a close proof-read. Having said this, I think the data merit publication, but I recommend a revision of the paper that addresses the points above. I wish the authors good luck and would be happy to review it again. Detailed comments follow below.

Response: We have added the weakness of cross-sectional study by design as suggested (Page 12 lines 444-446; 453-454). We have reworked on SEM section in the Statistical analyses (Page lines 286-288; 296-298). We have addressed the abbreviations and inconsistency issues with the tracked changes reflected in the main text. We have also added a flow chart for study design to make it clear

on inclusions and exclusions made.

Abstract:

2: Objectives: Abbreviate blood pressure; put BP in parentheses.

Response: Done as suggested.

3: Design: Cross-sectional instead of cross-section

Response: Done as suggested.

4: Results, line3: should it be "...effect of BMI on SBP".

Response: Done as suggested.

5: Results: Please write "systolic BP (SBP)" for proper abbreviation.

Response: Done as suggested.

6: Results, line6: use "BMI" (already abbreviated above)

Response: Done as suggested.

7: Conclusions, line2: use "SES" (already abbreviated above).

Response: Done as suggested.

Strengths and limitations bullet points:

8: Strength 1: Add "equation" between "structural" and "modelling".

Response: Done as suggested.

9: Strength 2: When harmonization is listed as a strength I wonder what kind of harmonization. Could the authors specify exactly what data that were harmonized?

Response: Done as suggested.

10: Limitation1: Suggest to delete "We are currently working on research..." from the bullet points as this is also mentioned in the main text.

Response: Done as suggested.

11: Limitation2: I suggest to exchange "precise" with "valid" or "accurate" as biases are not threatening precision, but validity/accuracy.

Response: Done as suggested.

12: Limitation3: Add "a" between "is" and "need" ?

Response: Done as suggested.

Introduction:

13: p5, para1: Abbreviate "blood pressure" to "blood pressure (BP)" at first mentioning or use just "blood pressure" throughout the ms.

Response: We have opted to use BP throughout the manuscript.

14: p5, para3: please define "SES" before abbreviation.

Response: Done as suggested.

Methods:

15: p6, study sample: It is unclear whether pregnant young women (n=51) were subtracted before or after you arrived at n=509 for the rural sample and n=510 for the urban sample. It is also unclear how many of the young pregnant women who were rural and how many who were

urban? Also, it is unclear how many who were excluded due to illness (last sentence). As mentioned in comment 1, I suggest adding a flow chart as a fig1 to clearly show the inclusions/exclusions made for both the rural and urban samples.

Response: We thank the reviewer for the comment and the proposal; we have rewritten and added a flow chart for the study design as reflected in Figure 1 in Methods section (Page 6 lines 222-225).

16: p6, blood pressure: Use of too large/small cuff-size has been associated erroneous BP readings. Were different cuff-sizes used to accommodate individual differences in overarm circumference?

Response: We had various cuff sizes and the appropriate size was used to accommodate differences in arm circumference, this has been added to the Methods section on Page 7 lines 247-248

17: p7, SES: Please write “SES” in first sentence (already abbreviated in intro)

Response: Done as suggested.

18: p8, statistical analyses: Please use “MVPA” (already defined)

Response: Done as suggested.

19: p8, statistical analyses: define and use “SBP” since you do not report DBP.

Response: Done as suggested.

20: p8, statistical analyses: As mentioned in comment 1, consider to give some more information on the interpretation of SEM – what do you estimate; β with 95% confidence intervals? Please also report significance level – 0.05?

Response: Thank you, we have edited to reflect what is being estimated and at what confidence intervals as reflected on Page 6 lines 296-298.

Results:

21: p8, study characteristics: I now understand that n=492 and n=476 were the numbers arrived at after excluding the 51 pregnant. However, I think it would be easier to follow with a flow chart as mentioned in comments 1 and 15.

Response: We have added a flow chart as in 1 and 15

22: p8, study characteristics: Just use MVPA (already defined)

Response: Done as suggested.

23: p9: In the heading you write out Structural equation models, but abbreviate BMI – perhaps use full wording in headings and abbreviations in text after first mention for consistency?

Response: Thank you, we have removed the abbreviations in the headings

24: p9, Structural equation models for BMI and waist circumference: Add “95% CI” when reporting from SEMs.

Response: Done as suggested.

Discussion:

25: p9, para2: use “SSA” instead of sub-Saharan Africa – already abbreviated.

Response: Done as suggested.

26: p10, para2, last sentence: Please report whether the direction of the association for muscle mass and bone mass on SBP was positive or negative in ref #63.

Response: We have written the sentence to make it clear, as reflected on Page 10 line 380, it is ref #65 now.

27: p10, para3: Add “a” between “At” and “population”.

Response: Done as suggested.

28: p10, para3: Be consistent in the use of “et al.” or “and colleagues”, and “pooled” or “combined” – choose one expression and stick with it throughout the ms as this is important for readability and clarity.

Response: We have opted to using colleagues instead of et al. and pooled instead of combined for consistency.

29: p11, para1: use either PA or physical activity consistently – in para4 “physical activity” is used again.

Response: We have opted to use PA.

30: p11, para2: here “combined” and not “pooled” is used, and “et al.” instead of “and colleagues”....

Response: Done as indicated in 28.

31: p11, para3: here “pooled” and not “combined” is used... Please excuse my hang-up, but I think it will improve the ms with more consistent use of expressions and abbreviations.

Response: Done as indicated in 28 and also used consistent expressions and abbreviations.

32: p11, para4, last sentence: As mentioned in comment 1, it is important that you elaborate here on the limitations inherent with a cross-sectional design; that you by design lack a temporal component between the factors analyzed. Thus, it is difficult to say anything certain about the direction of the associations.

Response: As in 1 we have added the weakness of cross-sectional design and suggested for a replication of our results in prospective data as reflected on Page 12 lines 445-447; 455

Conclusions:

33: p11-12: Based on comment 1 and 32 with respect to the limitations of the design, more prudence should be shown in the conclusions. You may consider to add something like: “Although our findings should be replicated with prospective data”

Response: Done, as reflected on Page 12 line 455.

Reviewer: 2

Reviewer Name: Séverine Sabia

Institution and Country: Inserm, France

Please state any competing interests: None declared

Please leave your comments for the authors below

This paper reports social disparities in blood pressure among rural and urban young women from South-Africa. It raises an interesting question, particularly in the context of transitions occurring in this country, and the authors nicely discuss their findings in relation to this context.

- 1. Analyses conducted separately among rural and urban women show interesting results both on SES disparities and on the associations between MVPA, BMI and blood**

pressure. However, results based on pooled analyses show confusing conclusion compared to stratified analyses. Is it possible that urban/rural environment is missing confounding factors in the pooled analysis? As household, BMI, physical activity and blood pressure differ strongly between urban and rural population, it would be nice to introduce this parameter into the model combining both populations.

Response: We thank the reviewer for the observation; the differences in SES results between pooled and individual sites have been pointed out in the Discussion section. We speculate that combining SES data between rural and urban introduced more heterogeneity within SES; this could therefore mean that individuals with more diverse SES levels expose different risks to SBP. In this regard at a population level, there is a need to consider different SES categories and monitor the effects of transitioning from one SES category to another, Page 11 lines 385 to 391. As the reviewer has pointed out we also do not rule out other factors such as dietary patterns which would be a rural/urban environment confounding factor, we did not have such data in the current study and it has been pointed out as one of the weaknesses of the current study.

2. Higher physical activity is associated with lower blood pressure in the combined analysis but not in the stratified analysis. The distribution of physical activity in the pooled sample is likely to be bimodal according to table 1. How do SEM deal with such distribution? Similar comment can be done for household assets.

Response: We have updated the Statistical methods section, (Page 9 lines 309-313), where we have added methods implemented in Stata version 14 and above to handle non-normal data SEM. We run the SEM with the Satorra–Bentler (implemented in STATA versions 14.0 and above) and Huber-White (Robust) Sandwich Estimator options [1]. These options relax the normality assumption hence robust to non-normal data [2, 3], which would be the case for mvpa and SES in the current study more especially after combining the rural and urban dataset. The estimates and goodness of fit indices were consistent in all of the options.

3. It would be nice to show association with waist circumference instead of BMI in appendix. Indeed in Table 1, we can see that rural participants have a lower BMI but a higher waist circumference (not significant). Showing in table 1, categories of waist circumference would be informative as well.

Response: We have added WC and education categories data in Table 1 as suggested. As indicated in the Results section, we observed similar results (direct and indirect effects) through BMI and WC. As in Table1 both categorized and continuous WC were not statistically difference between urban and rural hence we could not discuss more on non-significant WC results. We have presented the SEMs for BMI and WC in Figure S1 below, plus for use of education as a SES proxy education instead of SES index as requested in 4.

Figure S1; A: Structural equation model for SES, MVPA and **BMI** on SBP pooled; B: Structural equation model for SES, MVPA and **WC** on SBP pooled; C: Structural equation model for **education**, MVPA and BMI on SBP pooled.

4. It would be interesting to see association with alternative SES markers such as educational level. Is this information available?

Response: We have included education level data and run the SEM on pooled data, the effects are similar as shown in Figure S1 (A and C). The questionnaire used to determine the SES index is from the Griffiths et al. study. It is a validated tool and the literature acknowledges the value of using assets/commodities as an indicator socioeconomic status [4, 5]. Though education level has been commonly used as the proxy for SES, it would not be the best proxy for SES in developing countries. It has been argued that mother's education may just be a proxy for other dimensions of household SES [6]. In this study both SES index and education show similar effect (direct and indirect) effects on BP, Figure S1 (A and C).

5. The paper includes limitations that are well acknowledged in the discussion section, including the absence of information on diet patterns, the cross-sectional design and self-report of physical activity.

Response: Thank you.

1. FORMATTING AMENDMENTS (if any)

Required amendments will be listed here; please include these changes in your revised version:

-Authors must include a statement in the Methods section of the manuscript under the sub-heading 'Patient and Public Involvement'.

This should provide a brief response to the following questions:

-How was the development of the research question and outcome measures informed by patients' priorities, experience, and preferences?

-How did you involve patients in the design of this study?

-Were patients involved in the recruitment to and conduct of the study?

-How will the results be disseminated to study participants?

-For randomised controlled trials, was the burden of the intervention assessed by patients themselves?

-Patient advisers should also be thanked in the contributorship statement/acknowledgements.

Response: Patient and Public Involvement

No patients private or public were involved in this study, as it was a community population based.

References:

1. Williams R, Allison PD, Moral-Benito E: **Xtdpdml: Linear dynamic panel-data estimation using maximum likelihood and structural equation modeling**. In.; 2016.
2. Stata Corporation. 2017a. **Satorra–Bentler adjustments**. <https://www.stata.com/features/overview/sem-satorra-bentler/> . Last accessed 30th June, 2018.
3. Stata Corporation. 2017b. **Stata Structural Equation Modeling Reference Manual Release 15**. Stata Press: College Station, Texas 77845.
4. Griffiths PL, Rousham EK, Norris SA, Pettifor JM, Cameron N: **Socio-economic status and body composition outcomes in urban South African children**. *Arch Dis Child* 2008, **93**:862-867.

5. Filmer D, Scott K: **Assessing asset indices.** *Demography* 2012, **49**:359-392.
6. Stewart, T., and Sandile Simelane. "Are assets a valid proxy for income? an analysis of socioeconomic status and child mortality in South Africa." *Unpublished document*(2005).

VERSION 2 – REVIEW

REVIEWER	Jo S Stenehjem Cancer Registry of Norway, Dept of Reasearch
REVIEW RETURNED	13-Jul-2018

GENERAL COMMENTS	No further comments.
----------------------

REVIEWER	Sabia Séverine Inserm, France
REVIEW RETURNED	02-Oct-2018

GENERAL COMMENTS	The authors have addressed my comments. Could the results based on the method accounting for non-normality of the exposure and those using education and WC be presented in supplementary files? I could not find these results in the paper.
---

VERSION 2 – AUTHOR RESPONSE

Reviewer 1

Reviewer Name: Jo S Stenehjem

No further comments.

Response: Thank you.

Reviewer: 2

Reviewer Name: Sabia Séverine

The authors have addressed my comments. Could the results based on the method accounting for non-normality of the exposure and those using education and WC be presented in supplementary files? I could not find these results in the paper.

1. Results using education and WC

Response: We have presented the SEMs for BMI and WC in Figure S1 below, plus for the use of education as a SES proxy instead of SES.

Figure S1; A: Structural equation model for **SES**, MVPA and **BMI** on SBP pooled; B: Structural equation model for **SES**, MVPA and **WC** on SBP pooled; C: Structural equation model for **education**, MVPA and **BMI** on SBP pooled; D: Structural equation model for **education**, MVPA and **WC** on SBP pooled.

To run away from duplication of the results and presenting all the tables for the non-normality assumptions, we present only the path diagrams with estimates from the SEMs based on Satorra–Bentler (implemented in STATA versions 14.0 and above). This option relaxes the normality assumption hence robust to non-normal data.

Figure S2 - SEMs based on Satorra–Bentler; A: Structural equation model for **SES**, MVPA and **BMI** on SBP pooled; B: Structural equation model for **SES**, MVPA and **WC** on SBP pooled; C: Structural equation model for **education**, MVPA and **BMI** on SBP pooled; D: Structural equation model for **education**, MVPA and **WC** on SBP pooled.

We added the following in the main manuscript as reflected on Page 9 lines 303-305.

“Similar results were observed when including waist circumference as the body composition indicator as shown in the SEM path diagrams with estimates in Figure S1 (supplementary data). The results from the SEMs with the Satorra–Bentler adjustment option, accounting for non-normality of the exposure, are shown in Figure S2 (supplementary data).”